# Assessment of Resilience of Pistachio Agroecosystems in Rafsanjan Plain in Iran

**Fatemeh Darijani, Hadi Veisi * , Houman Liaghati, Mohammad Reza Nazari and Kours Khoshbakht**

Environmental Sciences Research Institute, Shahid Beheshti University, G.C., Evin, Tehran 1983963113, Iran; F_darijani@sbu.ac.ir (F.D.); h-liaghati@sbu.ac.ir (H.L.); mo_nazari@sbu.ac.ir (M.R.N.); kkhoshbakht@sbu.ac.ir (K.K.)
* Correspondence: h_veisi@sbu.ac.ir; Tel.: +98-21-2243-1971

**Abstract:** This study assessed the resilience of pistachio production systems in the Rafsanjan plain in Iran using an index of behavior-based indicators. One-hundred fifty pistachio orchards located in five major production areas were studied in 2016. The data was subjected to three-step multi-criteria analysis, including (i) normalization and aggregation; (ii) determination of the weights representing the priorities for each criterion and evaluation of the performance of each indicator; and (iii) comparison. The results showed that the study areas had problematic statuses regarding the indicators of membership in grassroots organizations, innate abilities, water sources, production stability, and insurance. They had critical or moderate statuses concerning the indicators of use of organic fertilizers, use of pesticides, soil fertility index, water-use efficiency (kg/m$^3$), trust in government, access to advisor services (extension), on-the-job training, and diversity of marketing. They had positive levels for the indicators of productivity, diversity of cultivars, diversity of on-farm practices, and exchange of information. We recommend the enhancement of the transformability capacity in PPSs by changing the focus from optimal states and the determinants of maximum sustainable yield (MSY paradigm) to adaptive resource management that includes developing participatory platforms for collaboration of usage of water resources.

**Keywords:** behavior-based indicators; pistachio orchard; resilience; regulation sources

## 1. Introduction

Over the last six decades, production practices in the agricultural sector of Iran's economy have been influenced by economic growth, land-use policies and population growth, and the resulting pressures. Between 1940 and 2010, the total urban population of Iran has increased from about 21% to 72%. Urbanization, industrialization, and intensification significantly affected soil and water resources [1] through mining, pollution, and increased effluent bearing heavy metals that degrade surface water quality.

The mean annual groundwater level in Iran over the past two decades decreased to 0.51 m. In 2008, the groundwater table decreased an average of 1.14 m. The average use of chemical fertilizers increased from about 2.1 million tons in the 1990s to about 3.7 million tons in 2009. Soil erosion also has totaled about 17 tons per hectare per year.

Alongside such challenges, a new, widely discussed concept of farm resilience emerged to meet the risks facing agriculture [2]. Resilience evokes evolutionary narratives, beginning with unexpected events, changes in transient effects, shocks, price volatility, uncertain market access, and complex sectorial policies. By integrating ecological resilience into the study of agroecosystems, it is possible

obtain valuable insight into agroecosystem identity, change, responsivity, and performance under stress [3].

Behavior-based indicators as a basic framework for resilience monitoring [4] were used to propose quantitative research approaches to tackle the continuing lack of biophysical and field-scale indicators needed to lend insight into dynamic resilience variables and mechanisms. The current study presents an index of behavior-based indicators that, when identified in an agroecosystem, suggest that it is resilient, and endowed with the capacity for adaptation and transformation. The indicators were compiled from characteristics of agroecosystems, such as the presence of these behavior-based indicators, to recognize resilience in an agroecosystem. Their absence or disappearance would suggest vulnerability and movement away from a state of resilience.

The ultimate goal is assessment of the resilience of pistachio agroecosystems. Pistachios are the most important agricultural product cultivated in tropical regions of Iran. Iran gains significant income from pistachio exports [5]. Given the many problems facing Iranian agriculture (drought, market fluctuations, water scarcity, etc.), it is necessary to improve the resilience of these systems. The current study aimed to estimate the resilience of pistachio farmers in the Rafsanjan plain in Iran against disturbance factors (drought, pests, and market) by adopting a set of indicators. The findings and results of this research could force policymakers and natural resource managers to focus on strengthening the resilience of pistachios in Rafsanjan plain by consideration of water conservation, sustainable production, and livelihood development.

## 1.1. The Concept of Resilience

Resilience emerged as a concept of unifying disciplines linked to sustainability [6]. Resilience theory was initially developed in the field of ecology, which is a multipurpose discipline and can be applied in a variety of ways [7,8]. The concept of resilience emerged as relating to complex and dynamic systems in economics [9], ecology [10], pedagogy [11], psychology [12], sociology [13], risk management [14], and network theory [15]. Resilience is the magnitude of disturbance that can be tolerated before a socioecological system (SES) enters a different region of the state space controlled by a different set of processes.

Resilience has several components that can be estimated in field studies. It is a component of sustainability [16], is characteristic of dynamic models, and comprises a sizeable proportion of SES [17]. SES is an important dimension of resilience. Natural resource management is not just related to ecological or social issues, but also to multiple integrated elements. Social and ecological systems have cultural, political, social, economic, ecological, and technological components that interact to focus on the "humans in nature" perspective where ecosystems are integrated into human society [18].

Studies on resilience show that the persistence of a social–ecological system led both to resistance to change and to change in the form of the system (adaptation) [8]. Resilience assessment tries to understand changing dynamics better. The practitioner's guide to resilience assessment, as created by the Resilience Alliance (RA), offers an iterative approach to understand how resilience, as a property of social-ecological systems, is created, maintained, or eroded over time [18]. There are two ecological and engineering perspectives in studying resilience. Engineering research focuses on resilience or robustness as recovery from perturbation, while ecological resilience focuses on adaptive capacity, which may lead to new equilibria [17].

## 1.2. Conceptualizing Resilience of Pistachio Agroecosystems in Rafsanjan Plain

Although the concept of resilience is valuable as a metaphor, one area in which it is critically underdeveloped is in metrics [17,19,20]. Because of its abstract and multidimensional nature, operationalization of resilience is challenging. Darnhofer et al. [4] claimed that models based on "resilience of what to what" (as suggested by Carpenter et al. [17]) cannot provide useful guidance to farmers because of complex and variable of farming systems over time and space. They suggested an

index of behavior-based indicators based on the rule of thumb for enabling farmers and facilitators to guide agricultural systems to a more resilient orientation.

Darnhofer et al. [4] confirmed that developing sets of surrogates or indicators is a better approach to assessing resilience. In agroecosystems, indicators of both individuals and groups are related to one of four phases in the adaptive cycle: Growth/exploitation, conservation, release, and reorganization/renewal [4,21,22]. The dimensions include ecological self-regulation, social self-organization, diversity, building of human capital, and economic dimensions for assessing resilience (see Section 3).

## 2. Materials and Methods

### 2.1. Study Area: Rafsanjan Plain

The Rafsanjan plain is located in Kerman province of Iran, (30°41″N, 55°99″) at an altitude of 1514 m. The city of Rafsanjan has a population of more than 280,000, making it the third largest city in the Kerman province (Figure 1). The Rafsanjan plain has an arid climate with hot summers and cold winters. An expanse of mountains surrounds the plain and greatly affects its climate. Most of the precipitation occurs in December through April. Rafsanjan is globally reputed for its high-quality pistachio production. Pistachio production and trade are the number one occupation in this region and pistachios are amongst the top agricultural commodities exported from Iran. The Rafsanjan plain comprises 2241 km$^2$ and includes the five major pistachio growing areas of Anar, Kashkoyeh, Nouq, Rafsanjan, and Kabootar-khan.

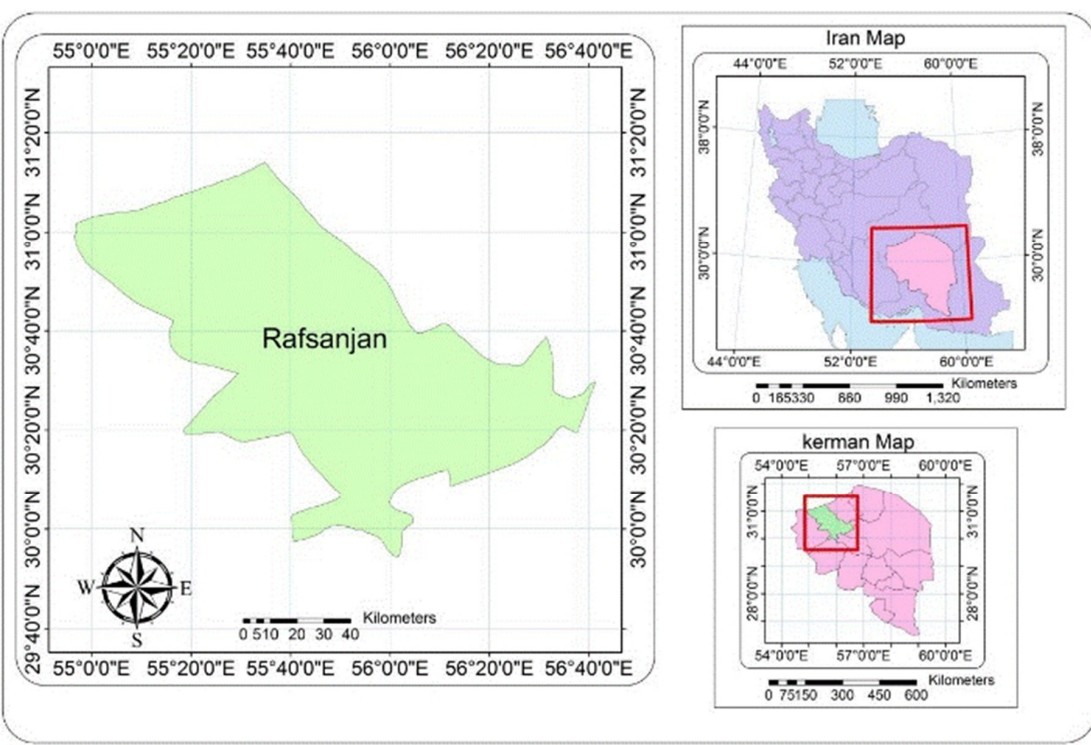

**Figure 1.** Study area map.

### 2.2. Study Methods

The data was collected in 2016 during face-to-face interviews using a questionnaire. Based on the farmer population, a 5% error rate from the mean and a 95% confidence interval (t ¼ 1.64), a sample size of 150 was elected by stratified sampling for the study location.

The questionnaire consisted of three major sections. The first section included information about personal characteristics and farm families. The second section covered the indicators associated with assessing resilience, which was derived from a study by Cabell and Oelofse [23]. Table 1 shows the sub-indicators from each group as distinguished from the main indicators, which were the basis for designing the items on the questionnaire. The third section was based on Ciftcioglu [24]. In this section, questions about the economy, society, and ecology were aggregated (using algebra and the linear summation method for indicators), and the framers were asked to rate each section on a scale of 1 to 3, 1 being least resilient and 3 being most resilient. The value of 1 (problematic; poor) was less than 40, 2 (critical, moderate) was 40–60 and 3 (positive, desirable) was over 60. The collected data was entered into spreadsheets and analyzed in SPSS (version 23). To calculate the indicators, a three-step process was carried out in multicriteria analysis [25]: Normalization and evaluation of the performance of each criterion (indicator), determination of the weights representing the priorities for each criterion, and aggregation (based on additive, multiplicative, or other distributional formalisms).

**Table 1.** The appropriate resilience assessment indicators for the pistachio production systems in the Rafsanjan plain.

| Indicators | Sub-Indicators (Variables) | Variables Scale | Reference |
|---|---|---|---|
| Ecologically Self-regulated | Use of organic fertilizers<br>Use of chemical fertilizers<br>Use of pesticides<br>Soil fertility index<br><br>Water-use efficiency | Amount of usage(kg/ha)<br>Amount of usage(kg/ha)<br>Amount of usage(kg/ha)<br>"Low," "Medium," "Optimum," and "Excessive" soil test categories<br>If no change = 0, if, 20% = 1, if 20–40 = 2, if .40% = 3 | [23,26–30] |
| Socially Self-regulated | Membership in Grass roots organizations<br><br>Degree of exchange of information<br><br>Degree of Job satisfaction<br><br>Level of access to trust in government<br><br>Level of access to advisor services (Extension) | If Not involved = 0, if involved = 1<br><br>If at least once a week = 3, if at least once a month = 2, at least once three months = 1, and lack of contact = 0<br>If unsatisfied = 0, satisfy = 1 (5 items) if not involved = 0, if involved = 1 (5 items)<br>if no trust = 0, if low trust = 1, if average trust = 2, if high trust = 3, and if very high trust = 4<br>If no access = 0, if very low access= 1, if low access = 2, if medium access = 3, if high access = 4, and if high access = 5 | [3,4,23,27,31] |
| Builds human capital | On-the-job training<br>Experience (capacity to work)<br>Innate abilities | Number of courses and workshops,<br>Year,<br>Three-point scale (for five items regarding their abilities in learning) | [23,32] |
| Diversity | Diversity of cultivated cultivars<br>Diversity of on-farm practices<br><br>Diversity of Marketing<br>Water from multiple sources | Number of cultivated cultivars<br>If done = 1, undone = 0 (regarding the doing of 10 practices in farmland)<br>Number of markets<br>Number of water resources | [7,26,33,34] |
| Economic | Insurance<br>Local investment<br><br>Production stability<br><br><br>Productivity | If No = 0, if Yes = 1<br>If Not use loan = 0, if use loan = 1,<br>Likert scale: If Likert rating scale: If very high = 4, high = 3, intermediate = 2 and low = 1<br>The ratio of results of the quantity of products output to input | [35] |

*(i) Normalization and Aggregation*

It is necessary to normalize the data because, during data collection, the measurement units of various indicators can differ [36]. The normalization method used in the current research was environmental performance indexing as developed by Esty et al. [37]. Accordingly, an additive

approach was applied in which the sub-indictors were summed and normalized into indicator classes (from 0 to 100). The highest value of 100 was the target and the lowest of zero was most undesirable condition.

*(ii)    Determination of Weights*

Principal component analysis (PCA) was used to estimate the relative weights in the overall sustainability measure [38,39], which is a crucial input to the aggregation process. The sub-indicators were weighted in proportion to the variance in the original set of variables as explained by the first principal component of that particular component [39].

*(iii)    Aggregation and Comparison*

After allocating weights to each sub-indicator and weighting the component scores, these scores are aggregated into a composite score. The mean and standard deviation for each index then were calculated to evaluate the resilience. Positive values denoted higher levels of resilience index (mean) and negative values denoted lower levels of resilience index (mean). The F-test was used to examine the differences in resilience among pistachio production systems in study area.

## 3. Results

### 3.1. Demographic and Farm Characteristics

Descriptive statistics for the socio-economic characteristics in Table 2 show that all of the farmers were males and illiterate, and their average age was 58 years. Over 60% of farmers had more than 20 years of farming experience and agriculture was their main job. Only 15 farmers had formal education. For family size (135 families), 90% had less than five members.

**Table 2.** Population profile of the farmers in the study area.

| Population Profile | Characteristics of the Population Profile | Number of Informants | Percentage of Informants |
|---|---|---|---|
| Gender | Male | 150 | 100 |
| | Female | 0 | 0 |
| Education | Primary school | 10 | 6.6 |
| | High school | 20 | 13.3 |
| | Illiterate | 120 | 80 |
| Employment | Farmer | 100 | 66.6 |
| | Retired | 50 | 33.3 |
| Age range | 30–39 | 25 | 16.6 |
| | 40–49 | 35 | 23.3 |
| | 50–59 | 90 | 60 |
| Years of farming experience | 0–19 | 15 | 10 |
| | 20–29 | 50 | 33.3 |
| | 30–39 | 95 | 63.3 |
| Family size | 1–2 | 40 | 26.6 |
| | 3–4 | 95 | 63.3 |
| | over 5 | 15 | 10 |

Evaluations also indicated that the average area under pistachio cultivation was 10.4 ha and the average yield was 1.9 tons/ha.

### 3.2. Ecological Self-Regulation

The resilience capacity of the pistachio production system (PPS) was assessed for its social, economic, and ecological dimensions. Self-regulating agroecosystems depend on the work of regulating

ecosystem services, such as biodiversity, the hydrological cycle, and soil resources [34–40]. To assess the resilience in the study area, the indicators were defined according to Cabell et al. [23]. To evaluate the ecological impact, as shown in Table 3, the use of organic and chemical fertilizers, pest management, soil fertility index, and water use efficiency were examined [3]. In this regard, the consumption of fertilizers, pesticides, and insecticides in the PPSs was very disturbing. For "use of pesticides," the resilience based on index value was problematic (< 40 to 100). Index values for "use of organic fertilizers," "soil fertility index," and "water use efficiency" ranged from 40 to 60, suggesting a critical or manageable level of resilience. Only the index value for "use of chemical fertilizers" was 61 (to 100) indicating positive resilience.

The highest index values for the use of chemical fertilizers were 67.5 kg in Noug county and 36.6 kg in Anar county for use of pesticide. However, there was no significant difference between the amounts of fertilizer and pesticide used in the five areas. These results revealed that the amount of fertilizer use is about 4000 kg ha$^{-1}$ in the study area, whereas it should be about 500 kg ha$^{-1}$. Pesticide use varied over the years and was affected by climate. The results also showed that the average use of organic fertilizer in Nouq and Rafsanjan was higher than in other regions, although there was no significant difference among regions.

Among the pistachio orchards, the average water use efficiency was 150 g/m$^3$ and the difference in water use efficiency between counties was significant. The index of water use efficiency of the pistachio orchards in Kabootar-khan and Rafsanjan counties was higher than in other counties. This finding revealed that the actual well water flows in Rafsanjan, Nouq, and Kashkoyeh were statistically similar as 19.61, 21.83, and 18.50 L/sec, respectively. Anar and Kabootar-khan, however, had water flows of 28.04 and 26.05 L/sec, respectively. The well water in Anar was more saline, whereas Kashkoyeh and Nouq had safer levels of salinity than Anar, which is consistent with previous reports [41]. This finding confirms that the water crisis is the main issue threatening resilience as well as the sustainability of pistachio production in the region studied.

*3.3. Social Self-Regulation*

Social resilience is the ability of groups or communities to cope with external stress and disturbances due to social, political, and environmental change. The resilience of a social system depends on ecological resilience because of the dependence of communities on ecosystems and their economic activities [13,42]. The Integrated Landscape Management Team states that the creation of cooperatives can affect social resilience [43]. On the other hand, strong horizontal and vertical social networks can enhance the adaptation capacity of social systems [13,44]. Therefore, cooperatives and institutions in the region can have a positive effect on the abilities of farmers to cope with stress events.

The resilience of the social system was assessed by membership in grassroots organizations, exchange of information, job satisfaction, trust in government, and extension. Considering the importance of a social index and the complex relationships between this index and others in a time of stress, sub-indicators were used to assess the social status of the regions in the study area (Table 4). The resilience values for "membership in grassroots organizations" (X = 39. 5, SD = 7.5) was < 40 and problematic, whereas "job satisfaction" (X = 53.6, SD = 16.32), "trust in government" (X = 44.1, SD = 11.45), and "advisor services (extension)" (X = 56.9, SD=19.7) scored 40–60 (moderate or manageable). "Exchange of information" (X = 64.3, SD = 14.2) scored >61, which was high and positive. About 75% of farmers are members of local cooperatives and periodically hold meetings with experts to train farmers on farming techniques and economic issues and where government policies are presented and discussed.

**Table 3.** The ecological resilience assessment indicators for the Rafsanjan plain.

| Indicators | Sub-Indicators | Mean | | | | | Mean | Sd | F Value | Sig |
|---|---|---|---|---|---|---|---|---|---|---|
| | | **Anar** | **Rafsanjan** | **Noug** | **Kashkoye** | **Kabootarkhan** | | | | |
| **Ecologically Self-regulated** | Use of organic fertilizers | 49.1 | 51.6 | 50.8 | 48.3 | 44.1 | 48.7 | 12.3 | 0.7 | 0.5 |
| | Use of chemical fertilizers | 57.5 | 59.1 | 67.5 | 61.6 | 61.4 | 61.4 | 21.6 | 1.6 | 0.1 |
| | Use of pesticides | 36.6 | 32.5 | 30.8 | 34.1 | 30.0 | 32.8 | 9.19 | 1.1 | 0.3 |
| | Soil Fertility Index | 45.8 | 50.0 | 45.0 | 51.0 | 49.1 | 48.1 | 14.2 | 0.9 | 0.4 |
| | Water-use efficiency | 46.2 | 55.0 | 42.3 | 31.6 | 59.13 | 46.8 | 16.1 | 1.82 | 0.04 * |

* Statistically significant at $p < 0.05$.

**Table 4.** The social resilience assessment indicators for the Rafsanjan plain.

| Indicators | Sub-Indicators | Mean | | | | | Mean | Sd | F Value | Sig |
|---|---|---|---|---|---|---|---|---|---|---|
| | | **Anar** | **Rafsanjan** | **Noug** | **Kashkoye** | **Kabootarkhan** | | | | |
| **Socially Self-regulated** | Membership in Grass roots organizations | 38.3 | 43.3 | 39.1 | 37.5 | 48.3 | 39.5 | 7.5 | 1.1 | 0.3 |
| | Exchange of information | 59.1 | 61.6 | 70.0 | 66.6 | 65.0 | 64.3 | 14.2 | 2.5 | 0.04 * |
| | Job satisfaction | 52.5 | 49.1 | 55.0 | 58.0 | 54.0 | 53.6 | 16.32 | 0.70 | 0.5 |
| | Trust in government | 43.3 | 45.8 | 46.0 | 41.6 | 45.8 | 44.1 | 11.45 | 0.9 | 0.4 |
| | Advisor services (Extension) | 55.6 | 57.6 | 58.1 | 56.3 | 54.0 | 56.9 | 19.7 | 0.05 | 0.9 |

* Statistically significant at $p < 0.05$.

The indicator "access to advisor services (extension)" scored higher in Noug County. Given the high potential for social participation in the region, it is hoped that the motivation and confidence of farmers in this area will increase. This is consistent with the results of Shahiki et al. [45].

The results indicated that there were no significant differences in job satisfaction, trust in government and access to advisor services (extension), but there were for the information exchange index. The difference in this index depended on the farmers' attitudes toward information exchange. For example, some farmers considered only experts and some of their friends to be credible, and this issue caused significant differences in the areas under study.

### 3.4. Building Human Capital

Building resilience of people and communities translates into building human capital that is ultimately better able to adapt to climate shocks and stressors [46]. The skill, ability, and knowledge of each person are, in fact, the assets at his disposal, and are the strongest capital and investment in the labor force. Becker [47] defined human capital as the stock of knowledge and skills that considers the contributions of human capital directly in the production process. Considering the importance of the topic, the main sub-indicators of on-the-job training, experience (capacity to work), and innate abilities can define human capital. Table 5 shows that there was no significant difference in these indices in the study areas. The indicator values for experience (capacity to work) was positive at >60. The values for on-the-job training and innate learning ability of an agricultural technique were 52.6 and 25.7 (to 100) at the critical and problematic levels, respectively.

Adamopoulos and Restuccia [48] interpreted innate ability for farming as farm-level productivity. They found that about 80% of farmers had attended workshops during the growing season (on-the-job training). However, young people were less interested in classes and did not consider agriculture to be a reliable occupation.

### 3.5. Diversity

This dimension suggests the heterogeneity of features both in the landscape and on the farm and diversity of inputs, outputs, income sources, markets, pest control practices, water sources, and irrigation methods. After the release phase of the adaptive cycle, it provides insurance against total system collapse and the seeds of renewal [21,49]. Cabell et al. [23] calculated the diversity of pistachio cultivars, farm inputs, variability of the sales market, water sources, and irrigation methods. Table 6 shows that there were no significant differences in the study area regarding indictors of diversity of cultivars and marketing. On average, two to three pistachio cultivars were planted in the study areas. In recent years, drought-tolerant and higher-yield cultivars were planted because of the water crisis.

Analysis of functional diversity revealed that the index of diversity for on-farm practices and water resources were significantly different in the study area with mean scores 78.46 and 39.1 (to 100), respectively. The index of diversity for on-farm practices (irrigation methods, pest management, etc.) in Rafsanjan, Noug and Kabootar-khan counties was higher than in Anar and Kashkoye counties. Diversity of water resources in Noug, Kashkoye, and Kabootar-khan counties was greater than for Noug and Anar counties. It appears that efforts to provide various responses for comparison in functional diversity were not useful.

**Table 5.** The Builds human capital resilience assessment indicators for the Rafsanjan plain.

| Indicators | Sub-Indicators | Mean | | | | | Mean | Sd | F Value | Sig |
|---|---|---|---|---|---|---|---|---|---|---|
| | | Anar | Rafsanjan | Noug | Kashkoye | Kabootarkhan | | | | |
| | On-the-job training | 53.3 | 50.0 | 53.3 | 50.0 | 56.6 | 52.6 | 15.3 | 0.09 | 0.9 |
| Builds human capital | Experience (capacity to work) | 66.6 | 75.0 | 73.3 | 78.0 | 74.0 | 73.3 | 19.2 | 0.20 | 0.8 |
| | Innate abilities | 26.6 | 25.8 | 26.9 | 23.3 | 26.0 | 25.7 | 5.1 | 0.10 | 0.9 |

**Table 6.** The diversity indicators for the Rafsanjan plain.

| Indicators | Sub-Indicators | Mean | | | | | Mean | Sd | F Value | Sig |
|---|---|---|---|---|---|---|---|---|---|---|
| | | Anar | Rafsanjan | Noug | Kashkoye | Kabootarkhan | | | | |
| | Diversity of cultivated cultivars | 61.1 | 66.6 | 85.7 | 45.4 | 75.0 | 66.86 | 12.6 | 1.7 | 0.1 |
| Diversity | Diversity of on-farm practices | 71.6 | 83.3 | 80.8 | 11.4 | 12.1 | 78.46 | 9.4 | 2.4 | 0.05 |
| | Diversity of Marketing | 63.3 | 62.5 | 65.8 | 11.8 | 18.8 | 67.82 | 15.3 | 1.1 | 0.3 |
| | Water from multiple sources | 31.3 | 34.6 | 44.8 | 42.5 | 42.3 | 39.1 | 12.50 | 1.75 | 0.04 * |

* Statistically significant at $p < 0.05$.



### 3.6. Economic Dimension

The economic dimension of resilience covers the following themes: Buffer mechanisms (savings, assets, insurance) to cope with uncontrolled change and shocks, production stability and production productivity [23,50]. The indicators of insurance, local investment, production stability, and productivity were used to evaluate the economic resilience of the study area (Table 7). The mean scores for insurance and production stability were < 40 (problematic), while for local investment and productivity were > 60 (positive). The results of the F-test revealed no significant differences between study areas regarding the economic indicators. However, pistachio orchards in Noug and Kashkoye scored higher than in the other counties for buffer mechanisms (local investment and insurance). The indictors related to production-stability and productivity showed that pistachio orchards in Noug, Kashkoye, and Kabootar-khan scored higher than in Anar and Rafsanjan. The productivity indicator was measured as the ratio of agricultural outputs to agricultural inputs and indicated that only the productivity of human labor was significantly higher between counties. Differences in the productivity of fertilizers, pesticides, insecticides, and machinery were not significant.

**Table 7.** The economical resilience assessment indicators for the Rafsanjan plain.

| Indicators | Sub-Indicators | Mean | | | | | Mean | Sd | F Value | Sig |
|---|---|---|---|---|---|---|---|---|---|---|
| | | Anar | Rafsanjan | Noug | Kashkoye | Kabootarkhan | | | | |
| Economic | Insurance | 20 | 13.3 | 22.0 | 23.0 | 22.3 | 19.72 | 4.72 | 1.20 | 0.12 |
| | Local investment | 73.3 | 70.0 | 76.0 | 74.0 | 71.0 | 72.86 | 23.12 | 0.10 | 0.9 |
| | Production stability | 10.0 | 16.6 | 20.0 | 18.0 | 19.0 | 16.72 | 7.21 | 1.40 | 0.09 |
| | Productivity | 63.3 | 64.0 | 65.0 | 68.0 | 66.0 | 65.26 | 17.8 | 0.03 | 0.9 |

## 4. Discussion

Applying an index of 21 indicators in five categories identified the resiliency of pistachio production systems and their capacity for robustness, adaptation, and transformation. The results of the evaluation found problematic status for the indicators of the use of pesticides, membership in grassroots organizations, innate abilities, water from multiple sources, production stability, and insurance. As Peterson et al. [3] asserted, the internal sources of regulation or internal buffering mechanisms in PPSs, particularly in Anar and Noug counties, were poor and the systems had been pushed into an undesirable state. However, internal regulations such as experience (capacity to work), local investment, diversity of on-farm practices and job satisfaction, especially in Kabootar-khan county, were positive, suggesting good potential for human capital to reorganize the PPSs.

Sinclair et al. [51] reported that the role of human agency is critical to understanding system resilience, which is the capacity of individuals to make sense of change. The results revealed that the indicators of the use of organic fertilizers and pesticides, the soil fertility index, water use efficiency $(g/m^3)$, trust in government, access to advisory services (extension), on-the-job training, and diversity of marketing are at the manageable (moderate) level. These results have suggested a platform for enhancing resiliency in PPSs in Rafsanjan.

Recognizing the influence of both internal and external regulation on the resilience of PPSs, we suggest a participatory platform [52] to bring together different stakeholders to identify solutions to their problems [53] regarding both regulatory sources and sustaining resilience as common goals. For example, the platform can contribute to forming a grass roots organization, which facilitates sharing of experiences, the risk of disaster and resources, particularly water resources, and drive the best management practices. The platform improves social and institutional learning by developing collaborative learning networks and promoting knowledge coproduction [18]. It also cooperates in collective efforts to eliminate or reduce the effects of shocks. This can be very effective when the appropriate status of the social indicators in the area studied is considered.

## 5. Conclusions

Resilience is an integrated framework from which to explore the ability of agroecosystems to cope with changing environments [54]. The abilities are robustness (a buffer that allow persistence), adaptability (the capacity to adjust responses to changing external drivers and internal processes), and transformability (the capacity to create a fundamentally new system when the existing system is untenable) [55,56]. We investigated the features of the pistachio production systems using behavior-based indicators of resilience in agroecosystems [4] at the problematic, manageable, and positive levels.

Given the relatively external nature of these manageable indicators, it can be argued that PPS management practices and processes have room for improvement. A shift toward a proactive approach is critical. It allows consideration of long-term goals and multiple services to maintain production when confronting environmental stress [3]. We found a high level of resilience for the indictors of productivity, diversity of cultivars, diversity of on-farm practices, and exchange of information, which can be considered a milestone for reorganization in PPSs in the adaptive cycle.

Given the nature of the information obtained about the PPSs of the Rafsanjan plain, it can be concluded that high external regulation (access to services, investment, agrochemicals, etc.) with somewhat low internal regulation (water resources, innate abilities, stability, etc.) resulted in low resilience and a higher likelihood of flipping to an undesirable (unproductive) regime. These findings verify the conclusions of Akbari et al. [57] and Peterson et al. [3], who articulated the internal and external regulations, e.g. the risk frequency of previous year (stability), diversity of cultivated crops, off-farm employment, crop acreage, participating in a crop supervisory, ownership of machinery, and insured crop acreage that affected Reducing Natural Disasters and Risk Management to Sustainable of PPSs. PPSs are at the reorganization step of an adaptive cycle and have a high capacity for transformability because the manageable level of a few indictors relate to the process of production. These change the focus from seeking optimal states and the determinants of maximum sustainable yield (MSY paradigm) to adaptive resource management. PPSs are robust because they have a high level of human capital (valuable farmer capacity and experience), local investment, and job satisfaction. PPSs have a proper level of adaptive capacity because of the diversity of cultivars, practices, and information sources. We suggest an increase in the transformability capacity in PPSs by altering the focus from searching for optimal states and the determinants of maximum sustainable yield (MSY paradigm) to adaptive resource management. This is the development of participatory platforms for collaboration in the use of water resources and promotion of the intrinsic ability of farmers as well as diversification of on-farm practices and technologies. This platforms can contribute to follow up the mechanisms and strategies that Razzaghi Borkhani et al. [58] addressed for Reducing Natural Disasters and Risk Management to Sustainable of Gardens include: "supportive-credit", "environmental-spatial", "socio-participation", "knowledge-awareness", "infrastructure-institutional", "educational-informational", and "economic factors" respectively.

Ultimately, although this study by integrating resilience into agroecosystems research provides a metrics for measuring resilience as well as some insights into the ability of agroecosystems, i.e. the productivity, stability, resistance, and recovery of system processes as a basic framework for resilience monitoring, there are still limitations to operationalize resilience thinking in agroecosystems [3]. Some of those limitations are in considering productive functions and cross-scale interactions. In this sense, more research is required to integrate the productive functions—by focusing on outcomes such as crop yield, farm income, and provision of ecosystem services—and understanding of multiple scales and speeds of influence both above and below the agroecosystem scale.

**Author Contributions:** All the authors equally contributed and commented on early and final version of the manuscript.

**Funding:** This research received no external funding.

**Acknowledgments:** This research was supported by Shahid Beheshti University, Iran. We thank mangers and experts in ministry of Jihad-for Agriculture, Iran for comments and contributions that greatly improved the research and provided insight and expertise that greatly assisted the research.

**Conflicts of Interest:** The authors declare no conflict of interest.

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
