# Peer review of "Assessment of Resilience of Pistachio Agroecosystems in Rafsanjan Plain in Iran"

_sustainability, doi:10.3390/su11061656_

Reviewer 1 Report

Moderate English changes required.

A few remarks:

Rows 20-23 (Abstract) - is the word "condition" most appropriate? Secondly - why "use of pesticides" is mentioned twice, in (i) and (ii)?

Row 33-34, similarly 36. Stating that agriculture "has been influenced" I would suggest to make clear to add in what way (direction) was agriculture affected. The same with row 36.

It seems soil erosion is an important issue. Was it covered by the the "soil-fertilility index"?

I wouldn't use  expression  "a synonim related to sustainability" as one of the meanings of the term Resilience (row 71). I agree that more resilient system should be more sustainable, but these are 2 different concepts.  Synonim, after Oxord Dictionaries", is "a word or phrase that means exactly or nearly the same", and in my opinion in the case of resilience and sustainability it is not the same.

Author Response

Reviewer(s)' Comments to Author:
Comment 1: Moderate English changes required.
Response: We have done thorough English editing and corrected the grammatical mistakes in the revised manuscript.
Comments to the Author Comment 2: Rows 20-23 (Abstract) - is the word "condition" most appropriate? Secondly - why "use of pesticides" is mentioned twice, in (i) and (ii)?
Response: Regarding these comments, “status” we have replaced with “condition” and have deleted the "use of pesticides" in (i) according to result.
Comment 3: Row 33-34, similarly 36. Stating that agriculture "has been influenced" I would suggest to make clear to add in what way (direction) was agriculture affected. The same with row 36.
Response: For these comments, we have provided more details to clarify directions of impacts. Comment 4: It seems soil erosion is an important issue. Was it covered by the the "soil-fertilility index"?
Response: Regarding this comment, the Soil Fertility Index (FI)’Schaetzl et al. (2012) is used as a family-level Soil Taxonomy information, i.e., interpretations of taxonomic features or properties. In this sense, a variety of indicators are used for assessing the state of erosion or soil fertility. For example, the potential erosion control of high, medium and low categories of erosion hazards was considered. Therefore, it can be argued that soil erosion is a key factor in the process of land degradation, loss of soil fertility and desertification.
Comment 5: I wouldn't use expression "a synonim related to sustainability" as one of the meanings of the term Resilience (row 71). I agree that more resilient system should be more sustainable, but these are 2 different concepts. Synonim, after Oxord Dictionaries", is "a word or phrase that means exactly or nearly the same", and in my opinion in the case of resilience and sustainability it is not the same.
Response: For this comment, in agreement with Marchese et al., (2018), we have defined resilience as a component of sustainability.

The following is a list of the peer review reports and author responses from the current submission.

Round  1

Reviewer 1 Report

I am satisfied with the final version.

Author Response

Dear Reviewer 1,

Thank you very much for the review of our manuscript entitled: “Assessment of resilience of pistachio agroecosystems in Rafsanjan plain in Iran”.

We are very much thankful to the reviewers for their deep and thorough review. We have done thorough English editing and corrected the grammatical mistakes in the revised manuscript. We hope that these revisions improve the paper such that you now deem it worthy of publication in sustainability, and our revision has improved the paper to a level of your satisfaction.  

Yours sincerely,

Hadi Veisi

Reviewer 2 Report

General points

There was  severe drought in 2016 in Kerman. In this year, the drought was caused to reduce harvest      of production   and to hollow the inside of pistachio. However, the total amount of      pistachio production didn’t change in Iran. Because production in other      provinces such as Yazd, North Khorasan and Semnan had compensated this      shortage. But according to the year and region that were selected in this study,      the results could not be generalizable.       

 In conclusion      section, it could be better, the results of this study compare with other      studies in Iran or out of Iran. Because there are many studies in Iran      that have worked on pistachio and have used each indicator separately.

In      conclusion section, it should be mentioned some suggestions for other      researchers and weaknesses or constraints of this study. 

Other points

1.     Table 1:

Some important indicators that could be considered

                                    I.          Climate changes such as sudden heat and cold. Pistachio is one of the most important products that is very sensitive to temperature.

                                  II.          Principles in harvesting, modifying from traditional to modern. One of the most important issue about Iranian pistachio is aflatoxin that is directly related to method to harvest this product.

                               III.          Price and whatever related to change the price.

                               IV.          Exchange rate and fluctuation of exchange rate that has influence on price and other economic factors.

                                 V.          Subsidy to input. According to new condition for paying subsidy, if removing subsidy from inputs are considered, productivity and local investment results are changed.

                                VI.          Instead of insurance, to use production risk, price risk, technological risk, labour risk, financial risk, instructural risk and damage risk could be more efficient. For all these, some papers that have worked in Iran could been referred.

2.     Table 2:

Gender: Total number of samples in this study is 150. According to the social condition of work in Iran and the culture in such a religious province like Kerman, it is strange that all the farmers for pistachio are women!

Employment: It could be important to mention that farming is their only job, or they have some other activities along farming.

Year of farming experience: why the range was started from10-19? It could be interesting to figure out the descriptive information about employment, experience and education with together. Is there any farmer with low level of education and high experience? Without comparison table, this information is just like numbers.

3. Table 3: the value of mean should be revise for water use efficiency.

4. Table 5: the value of mean should be revise for local investment and productivity.

Author Response

Response to reviewer comments

General points

There was severe drought in 2016 in Kerman. In this year, the drought was caused to reduce harvest of production and to hollow the inside of pistachio. However, the total amount of pistachio production didn’t change in Iran. Because production in other provinces such as Yazd, North Khorasan and Semnan had compensated this shortage. But according to the year and region that were selected in this study, the results could not be generalizable.

Ø  Regarding this comment, we agree to reviewer that the results could not be generalizable. However, our study has been done in last year and our focus was on assessment of resilience status at last year in Rafsanjan Plain. Therefore, we did not claim the resilience status of PPSs in Iran. This study is also unique in terms of developing a metrics for assessing resilience in agroecosystems.

In conclusion section, it could be better, the results of this study compare with other studies in Iran or out of Iran. Because there are many studies in Iran that have worked on pistachio and have used each indicator separately.

For this comment, we agree to reviewer, and comparison the results with other studies has been done. 

In      conclusion section, it should be mentioned some suggestions for other researchers and weaknesses or constraints of this study. 

For this comment, we have added a paragraph, in which, represents suggestions for other researchers and weaknesses or constraints of this study. 

Ø  Regarding these comments, the section conclusion has been revised and two paragraphs added to compare the results with other studies, and to represent suggestions for other researchers and weaknesses or constraints of this study. 

Other points

1.     Table 1:

Some important indicators that could be considered

                                    I.          Climate changes such as sudden heat and cold. Pistachio is one of the most important products that is very sensitive to temperature.

                                  II.          Principles in harvesting, modifying from traditional to modern. One of the most important issue about Iranian pistachio is aflatoxin that is directly related to method to harvest this product.

                               III.          Price and whatever related to change the price.

                               IV.          Exchange rate and fluctuation of exchange rate that has influence on price and other economic factors.

                                 V.          Subsidy to input. According to new condition for paying subsidy, if removing subsidy from inputs are considered, productivity and local investment results are changed.

                                VI.          Instead of insurance, to use production risk, price risk, technological risk, labour risk, financial risk, instructural risk and damage risk could be more efficient. For all these, some papers that have worked in Iran could been referred.

Ø  Regarding these comments, reviewer has mentioned invaluable comments that can be consider in resilience studies.  However, as mentioned in manuscript, we applied behavior-based indicators that were compiled from characteristics of agroecosystems for this study according to Darnhofer et al. (2010). It is notable that most of indicators and variables mentioned by reviewer are embedded in our indicators or stated in other words. 

2.     Table 2

Gender: Total number of samples in this study is 150. According to the social condition of work in Iran and the culture in such a religious province like Kerman, it is strange that all the farmers for pistachio are women!

For this comment, male was correct, and the sentence has been amended according to the comment.

Employment: It could be important to mention that farming is their only job, or they have some other activities along farming.

For this comment, we revised senesce according to reviewer suggestion. 

Year of farming experience: why the range was started from10-19? It could be interesting to figure out the descriptive information about employment, experience and education with together. Is there any farmer with low level of education and high experience? Without comparison table, this information is just like numbers.

Ø  The variable classified from 0-19 to clarify the point raised by the reviewer. 

3. Table 3: the value of mean should be revise for water use efficiency.

4. Table 5: the value of mean should be revise for local investment and productivity

Ø  For this comments, both table 3 and 5 have been checked and corrections have been done